# Seeing oneself as a data reuser: How subjectification activates the drivers of data reuse in science

**Marcel LaFlamme**[1]*, **Marion Poetz**[2,3], **Daniel Spichtinger**[4]

**1** Public Library of Science, San Francisco, California, United States of America, **2** Department of Strategy and Innovation, Copenhagen Business School, Copenhagen, Denmark, **3** Open Innovation in Science Center, Ludwig Boltzmann Gesellschaft, Vienna, Austria, **4** Bereichsleitung, Ludwig Boltzmann Gesellschaft, Vienna, Austria

* mlaflamme@plos.org

## Abstract

Considerable resources are being invested in strategies to facilitate the sharing of data across domains, with the aim of addressing inefficiencies and biases in scientific research and unlocking potential for science-based innovation. Still, we know too little about what determines whether scientific researchers actually make use of the unprecedented volume of data being shared. This study characterizes the factors influencing researcher data reuse in terms of their relationship to a specific research project, and introduces subjectification as the mechanism by which these influencing factors are activated. Based on our analysis of semi-structured interviews with a purposive sample of 24 data reusers and intermediaries, we find that while both project-independent and project-dependent factors may have a direct effect on a single instance of data reuse, they have an indirect effect on recurring data reuse as mediated by subjectification. We integrate our findings into a model of recurring data reuse behavior that presents subjectification as the mechanism by which influencing factors are activated in a propensity to engage in data reuse. Our findings hold scientific implications for the theorization of researcher data reuse, as well as practical implications around the role of settings for subjectification in bringing about and sustaining changes in researcher behavior.

## Introduction

The value of scientific data is increasingly assessed in terms of "the extent to which they are mobilized across contexts and aggregated with others" [1]. Widespread efforts are underway on the part of policymakers, funders, and other actors to promote data sharing in the name of making scientific research more efficient, reliable, and impactful [2–4]. Evidence suggests that these efforts are bearing fruit: perceptions of data sharing among scientists have grown more favorable over the past decade [5]. And while private or controlled forms of sharing are still the norm in many fields [6, 7], the open science movement has advanced an influential vision of making research data accessible to everyone with as few restrictions as possible [8]. Yet with

**Data Availability Statement:** Deidentified research data are available from the Austrian Social Science Data Archive (https://doi.org/10.11587/9ZDCYI).

**Funding:** This work was supported by the Austrian National Foundation for Research, Technology, and

Development (http://www.stiftung-fte.at). Funding was awarded to MP as grants for the Open Innovation in Science Center. The funder had no role in study design, data collection and analysis, decision to publish, or preparation of the manuscript.

**Competing interests:** One of the authors (ML) is at the time of publication an employee of PLOS, which publishes PLOS ONE. The remaining authors have no competing interests to declare.

all of the resources now being mobilized to realize this ambition, a nagging question remains [9]: even if data are widely shared, how do we know whether they will actually be reused and, thereby, yield additional value to both science and society?

A distinct literature on data reuse has begun to emerge, including conceptual work on just what it comprises. Scholars have documented often iterative stages of data reuse, including search and retrieval [10], sensemaking [11], and reprocessing or repurposing [12]. Others have offered typologies of approaches to reuse, as with one framework of a continuum from (more common) comparative to (less common) integrative reuses [13]. A meta-analysis of data reuse definitions identified the character of the data, the user, the purpose of use, and the time elapsed between production and use as essential criteria, each of which curiously confounds any firm distinction between use and reuse [14]. Indeed, the complexity and heterogeneity of data reuse practices make it difficult to do something as basic as measuring their incidence; even as formal data citation becomes more common, there are varieties of reuse that never get cited and thus never get counted [15]. Researchers also reuse data generated for scientific purposes as well as other types of public, proprietary, and personal data, which are produced, managed, and governed in very different ways [16].

Numerous empirical studies have sought to identify factors that facilitate or block researcher data reuse. A systematic review of these studies [17] pointed to individual-level factors such as the researcher's background and motivation, institutional-level factors such as requirements and facilitating conditions, and factors related to the data under consideration. The review also noted that the body of work it examined was relatively underdeveloped in terms of its application and elaboration of theory. Many of the studies reviewed did not engage with theory at all or relied on cognitivist approaches like the theory of planned behavior [18], which use constructs derived from individual beliefs to predict behavioral intentions but do not explain how those beliefs form or become salient. Meanwhile, sociomaterial approaches like the knowledge infrastructures research program [19] offer an account of the interaction between epistemic, social, and technical structures, but have little to say about the individual actor who navigates them. The disconnect between these lines of inquiry limits our understanding of how factors influencing researcher data reuse exert their effects over time and across levels of analysis. Accordingly, this study asks: what is the mechanism by which factors influencing data reuse are activated in a propensity to engage in reuse behavior?

To answer this research question, we draw on semi-structured interviews with a purposive sample of 24 data reusers and intermediaries from a range of disciplines and types of organization. Taking into account existing insight on factors influencing reuse behavior, we inductively explored mechanisms by which these factors give rise to a propensity to engage in data reuse: that is, to engage in reuse on a recurring basis rather than as a single instance (see [20] on propensity and frequency). Over multiple rounds of coding and categorizing our data, we found that issues of self-understanding are linked to the formation of a propensity to engage in reuse behavior. We then validated this link between self-understanding and recurring reuse, drawing on written follow-up responses from 8 of the interviewed reusers. We integrate our findings into a model of recurring data reuse behavior that introduces *subjectification* as a mediating mechanism by which influencing factors are activated in a propensity to engage in data reuse. Our results extend existing theoretical approaches to data reuse, by proposing scripts of scientific selfhood previously discussed in terms of persona and identity [21, 22] as a point of articulation between different types of factors influencing reuse behavior at the individual, project, and institutional levels of analysis. Furthermore, we connect an emerging stream of literature on scientific subjectification [23] to a new empirical context of data reuse. More broadly, we advance a multilevel program of research on the antecedents, contingencies, and consequences of the many varieties of openness and collaboration in science [24], of which data reuse is just

one example. Our findings also carry relevant practical implications. Policymakers and research organizations should be advised that technical solutions may not lead to a lasting change in researcher behavior without creating settings for subjectification that allow for the formation of new self-understandings compatible with the practice of data reuse.

In what follows, we briefly review existing knowledge about the factors influencing researcher data reuse and the theoretical approaches that underpin it. Next, we describe the process of selecting, collecting, and analyzing our data. We then present our findings about the mechanism by which factors influencing data reuse exert their effects on recurring reuse behavior, as well as the nature of the distinction between project-independent and project-dependent influencing factors. Finally, we discuss the implications of our findings for both theory and practice, as well as the study's limitations and avenues for future research.

## Data reuse in science: The state of existing knowledge

### Factors influencing researcher data reuse

Individual-level factors influencing whether researchers engage in data reuse can be highly idiosyncratic, reflecting personal investments in existing datasets and motivations for turning to them at a particular moment in scientific careers and lives [25]. However, studies have found that personal drivers may be less important for data reuse than they are for data sharing, suggesting that reuse behavior tends to be pragmatic rather than aspirational [26]. On this view, reuse is motivated by expectations that reusing data will be less costly in terms of time, money, and effort than collecting new data [27] and will allow the reuser to achieve their self-defined goals [28]. Conversely, when existing data will not allow researchers to answer the research question that they have previously formulated, then reuse is unlikely to take place [29]. Such judgments are moderated by researchers' assessments of their own abilities and past experiences: thus, for instance, failed reuse experiences may serve as a barrier to subsequent efforts at reuse [30].

Institutional-level factors influencing whether researchers engage in data reuse reflect the material conditions of how different scientific fields produce data, such as their reliance on centralized versus distributed instrumentation, as well as social practices that accrete over time into normative ways of working [31]. Such norms can undergo rapid change triggered by exogenous developments, as when moves to patent the sequence of the human genome in the early 1990s led to the formalization of processes for the reuse of genomics data [32]. But there has been little appetite among policymakers for mandating data reuse as such, which is generally viewed as an infringement on academic freedom in a way that mandating data sharing is not. Requirements around data reuse instead relate to narrower issues of compliance with legal restrictions [33] and ethical obligations [34], which can block certain forms of reuse. Meanwhile, investments in curation by data intermediaries like repositories have been shown to initiate virtuous cycles of reuse [35].

A systematic review [17] of empirical research on both data sharing and reuse identified eleven categories of factors influencing whether individual researchers engage in these practices: the researcher's background, the researcher's experience and skills, personal drivers and intrinsic motivations, effort, expected performance, trust, social influence and affiliation, facilitating conditions, requirements and formal obligations, legislation and regulations, and data characteristics. However, the review did not propose to structure these categories or to explore their interactions; all eleven are posited to have direct effects on data reuse, with little discussion of mediating or moderating relationships, weighting of different influences, or processual dynamics. At one point in the review, the authors call for "the development of a new theory, for which the categories and factors derived from our thematic analysis can be used as a basis."

Such a theory would likely need to clarify the relationship between different types of factors and levels of analysis, and to elucidate the mechanisms by which influencing factors are activated.

## Theoretical approaches to researcher data reuse

The systematic review discussed above [17] indicates that the existing theoretical framework most widely applied in the empirical studies it analyzes is the theory of planned behavior, as formulated by the social psychologist Icek Ajzen [18]. This individual-level approach aims to predict and explain behavioral intentions on the basis of three constructs: *attitudes* toward the behavior, understood as the perceived likelihood of its performance leading to a particular outcome or experience; *norms* about the behavior, understood as the perceived likelihood that other social actors approve of the behavior and engage in it themselves; and *perceived behavioral control*, or the degree to which the individual understands themselves to be capable of and free to perform the behavior. In one study of health scientists' data reuse behaviors [36], researchers operationalized these three constructs in a fixed-response survey and found strong support for them as predictors of an intention to reuse. Yet because the theory of planned behavior does not offer an account of how the beliefs from which attitudes and norms are said to follow form in the first place [37], the theory is of limited value in explaining how researchers come to hold and assign particular weights to their beliefs about reuse. The tenuousness of the link between intention and actual behavior is another limitation of the theory, as Ajzen and his longtime collaborator Martin Fishbein [38] have acknowledged. Studies of data reuse have, accordingly, extended the theory of planned behavior with other constructs to account for institutional-level factors that influence whether or not reuse takes place. Yet such ostensibly multilevel studies [39] tend not to specify how factors at different levels of analysis interact, showing cross-level effects in their statistical analysis but passing over the question of the mechanism by which individual and institutional factors are joined up.

Another theoretical approach not mentioned in the systematic review, but found to be significant in our own review of the literature on data reuse, is the knowledge infrastructures research program [19]. In contrast to the theory of planned behavior, which stakes its explanatory power on a small number of precisely defined constructs, the knowledge infrastructures research program tends to be more capacious in its conceptualization. For instance, the information studies scholar Christine Borgman has used the term *knowledge infrastructure* to refer to "an ecology of people, practices, technologies, institutions, material objects, and relationships" [31]. While an openness to factors influencing behavior at multiple levels of analysis and an emphasis on their interconnection are useful features of this approach, it can be difficult to determine where the influence of such an expansive construct starts and stops. Empirical studies of data reuse carried out within the knowledge infrastructures research program often emphasize the facilitating role of technical standards and the expert labor of data professionals but take a cautious, particularist approach to characterizing reusers themselves [40]. Recently, scholars have expressed concerns about how this orientation toward institutions may result in "a managerial understanding of knowledge infrastructures, which obscures effects on the people who are on the receiving end of knowledge infrastructure capacity" [41].

Even as the prevailing theoretical approaches to explaining data reuse focus on either individual or institutional levels of analysis, historical and social studies of science are increasingly concerned with issues of self-understanding that cut across these levels. Discussions of scholarly personae, or "models embodying the personal attributes that are regarded as necessary for being a scholar" [21], have allowed researchers to pursue questions around how such models are transmitted and reworked over time. Meanwhile, contemporary transformations of the

political, economic, and cultural context in which research organizations operate have shaped the very process by which researchers construct academic identities [22]. Against this back-drop, another concept that has shown promise for linking individual and institutional levels of analysis is that of subjectification. For the critical theorist Michel Foucault [42], subjectification refers to the process whereby a particular kind of socially legible person comes into being. This process both enables and constrains the individual undergoing it, who at once experiences themselves as the *subject of* their own thoughts and desires and is *subjected to* a certain shaping of what is thinkable or desirable. Recently, the science studies scholar Lisa Sigl [23] has defined subjectification as "a non-deterministic process by which actors come to understand themselves in relation to their particular social, cultural, and institutional environments," and has called for the study of subjectification as a site where structural transformations in the science system are registered and translated into modest, but consequential changes in individual research practices. While issues of self-understanding have not as yet been empirically investigated in relation to data reuse, concepts such as persona, identity, and subjectification may offer some traction on the problem of how different types of factors known to influence reuse behavior at different levels of analysis actually exert their effects.

## Methods

This study takes an exploratory qualitative approach to identifying the mechanism by which factors influencing data reuse by scientific researchers are activated in a propensity to engage in reuse behavior. That is, the study does not seek to test the applicability of a predefined theoretical framework, nor does it advance knowledge claims by measuring the covariance of fixed constructs. Instead, it aims to elicit the perspectives of study participants in words of their own choosing, and to discover patterns that shed light on social phenomena that are not yet adequately understood [43]. The interpretation of these patterns, in turn, forms the basis for novel conceptualizations that may subsequently be operationalized and validated using other empirical methods [44].

The *selection* of participants for this study followed a purposive sampling strategy [45], aimed at eliciting first-order perspectives from researchers who self-reported that they had previously engaged in data reuse as well as second-order perspectives from intermediaries who, in various capacities, had worked to facilitate data reuse by researchers and so could describe their perceptions (veridical or otherwise) of the factors influencing researcher data reuse. In contrast to studies of data reuse that emphasize attunement to particular "data cultures" [46], we aimed to maximize variation with respect to attributes like research field so as to be able to advance a more general set of claims. At the time the study was conducted, our institutions did not require approval for this type of study by an institutional review board or ethics committee. We made initial contacts through the professional networks of the research team; these contacts, in turn, were invited to recommend other prospective participants, a process that continued until preliminary analysis showed that data saturation had been reached [47]. In the end, our study population comprised 24 participants, including 12 researchers and 12 intermediaries. Table 1 provides an overview of their characteristics.

The *collection* of data for this study took place in two phases. In Phase 1, we collected data via semi-structured oral interviews [48], which were organized around parallel interview guides informed by our review of the literature on data reuse (see S1 File). For both researchers and intermediaries, our questions first addressed the nature of their experience with (facilitating) data reuse before turning to influencing factors. All questions were initially formulated in an open-ended way (e.g., researchers were asked "What helped or hindered you in your reuse of data?"), allowing participants to provide us with their own thoughts and experiences. After

**Table 1. Characteristics of the study participants.**

| Participant | Category | Gender | Researcher field or intermediary type | Position | Country | Follow-up response |
|---|---|---|---|---|---|---|
| 1 | Researcher | F | Economics and business | Postdoctoral researcher | Austria | Yes |
| 2 | Researcher | F | Health sciences | Assistant professor | Austria | Yes |
| 3 | Intermediary | M | Data center | Scientific officer | Belgium | N/A |
| 4 | Researcher | M | Sociology | Postdoctoral researcher | Austria | Yes |
| 5 | Researcher | F | Economics and business | Postdoctoral researcher | Australia | No |
| 6 | Intermediary | F | Data center | Director | France | N/A |
| 7 | Intermediary | M, M | Data center | Director, digital archivist | United States | N/A |
| 8 | Intermediary | M | Clinical evidence base | Director | Germany | N/A |
| 9 | Intermediary | F | Data services provider | Managing director | Germany | N/A |
| 10 | Intermediary | M | University | President | Netherlands | N/A |
| 11 | Researcher | F | Economics and business | Assistant professor | Denmark | Yes |
| 12 | Researcher | M | Languages and literature | PhD student | Netherlands | Yes |
| 13 | Researcher | M | Environmental sciences | Postdoctoral researcher | France | No |
| 14 | Intermediary | M | Data journal | Executive editor | China | N/A |
| 15 | Intermediary | M | Research institute | Open data officer | Germany | N/A |
| 16 | Researcher | F | Agricultural biotechnology | Associate professor | Bulgaria | Yes |
| 17 | Intermediary | F | Data center | Data curator | United Kingdom | N/A |
| 18 | Intermediary | M, F, M | University library | Digital library manager, officer, specialist | South Africa | N/A |
| 19 | Intermediary | M | Research funder | Program director | United Kingdom | N/A |
| 20 | Researcher | M | Clinical medicine | Group leader | Germany | Yes |
| 21 | Intermediary | M | Scientific publisher | Program director | United Kingdom | N/A |
| 22 | Researcher | M | Environmental sciences | Department head | Germany | No |
| 23 | Researcher | M | Other humanities | Associate professor | United States | Yes |
| 24 | Researcher | M | Physical sciences | Scientific staff | Switzerland | No |

Researcher fields were assigned based on the OECD Revised Field of Science and Technology Classification. Intermediary types were assigned based on the organization where the participant was employed at the time of the interview and the role that the organization plays in facilitating researcher data reuse.

the participant had completed their initial response, we probed the relevance of potential responses identified in our review of existing research on data reuse that the participant had not mentioned. Given that little is known about the mechanism by which such factors are activated, the interview guides did not include any direct questions about activating mechanisms. Phase 1 interviews were conducted from July through September 2019 and lasted an average of 40 minutes. Participants were given the option of conducting the interview in either English or German, and oral consent was obtained at the outset of each interview. Interviews took place either in person or over a video call, depending on the participant's location. With the consent of the participants, all interviews were audiorecorded and subsequently transcribed. Taken together, the interviews from Phase 1 resulted in a corpus of 286 pages of transcripts or 16 hours of audio. In September 2020, all participants were recontacted to ask for consent to deidentify their interview transcript and share it for purposes of scientific reuse; all but three gave their consent. In Phase 2 of data collection, which took place in October and November 2021, we recontacted the 12 researchers in our sample and invited them to provide a written response to a series of follow-up questions [49] seeking to validate the activating mechanism for recurring data reuse that we inductively identified in Phase 1. Of the 12 researchers in our sample, 8 completed the online form that we created to capture their responses (see S2 File).

The *analysis* of our data in Phase 1 involved multiple rounds of coding and categorizing, including iterative processes of making codes, organizing to code, and putting patterns

together [50], with a view to a) characterizing the factors that influence researcher data reuse in a meaningful way and b) identifying the mechanism by which these factors are activated in a propensity to engage in data reuse. In an initial round, two coders (one coauthor, plus one postdoctoral researcher not otherwise involved with the study) analyzed each transcript independently, before comparing and contrasting the codes they had assigned. This process was undertaken not only to mitigate against bias, but to reflexively explore differences in interpretation as inputs to the work of conceptualization [51]. In subsequent rounds, we used the qualitative data analysis software Atlas.ti to add, remove, merge, and split codes as the structure of our findings began to emerge. As a robustness check, we recoded the transcripts with the final codebook (see S3 File) to ensure that the constructs we had identified fit closely with the data. Data generated in Phase 2 were then deductively analyzed using the constructs identified in Phase 1, following the same process of independent analysis and coding comparison.

## Results

While our analysis confirms the role of a number of influencing factors identified in existing scholarship on researcher data reuse, it offers new insight in two respects. First, it characterizes these influencing factors in terms of their relationship to a specific research project involving the prospective reuser. Second, it shows that while these influencing factors may have a direct effect on a single instance of data reuse (for example, in the context of a onetime course assignment [Participants 4, 23]), they have an indirect effect on recurring reuse as mediated by the activating mechanism of subjectification. For the purposes of this study, we define subjectification as a multilevel process whereby researchers come to see themselves as data reusers and intermediaries seek to bring about this self-understanding on the part of researchers. We integrate our findings into a model of recurring data reuse behavior (Fig 1) that presents subjectification as the mechanism by which influencing factors are activated in a propensity to engage in data reuse.

In what follows, we discuss each element of the model in turn and provide evidence for the arguments that it advances. All quotations from the oral interviews are presented in English; translations from the German-language interviews were checked by a native speaker.

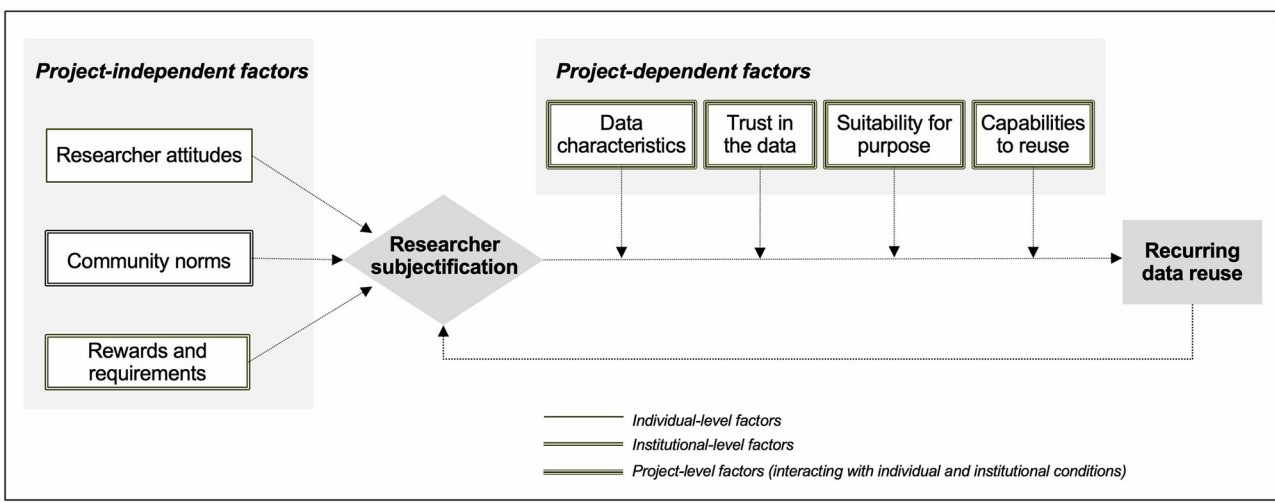

**Fig 1. A model of recurring data reuse behavior.**

## Subjectification as activating mechanism

Subjectification was initially identified as a pattern in the oral interviews, with 6 out of 12 researchers and 6 out of 12 intermediaries making statements associated with it. Then, this pattern and its relationship to recurring reuse behavior were validated with the written follow-up responses. Of the 8 researchers who provided a follow-up response, 5 indicated that they saw themselves as data reusers while 3 indicated that they did not (even though the latter had each engaged in data reuse on at least one occasion). As we discuss below, these results closely correspond to the coding of the oral interviews for subjectification.

In the oral interviews, study participants described the process of coming to see oneself as a data reuser with a variety of constructs, which we inductively identified and then sought to bring together in one higher-order concept. For instance, researchers referred to the development of a data reuse *mindset*, an approach to research design that takes data reuse to be an option worth considering for any project (compare [52]). One researcher (Participant 11) reflected: "I think that for newcomers, like PhD students or postdocs, typically the first thought is like, 'Let's collect the data on my own.' I think this might be something important. . . [instead] developing a mindset that, 'Yeah, we could just reuse data, we don't have to collect the data from scratch." Another (Participant 4) admitted that, although he reuses data in his teaching, when it comes to his own research "I notice. . .that I am of course also in this automated mode, I collect my data and so on." Here, drawing on a control systems metaphor, this researcher reflects on the incongruity of a mindset that embraces reuse in one context but fails to even consider it in another.

For their part, intermediaries referred to the process of coming to see oneself as a data reuser in terms of *awareness*. While we initially interpreted this term narrowly, variation in its usage across multiple interviews prompted us to realize that it referred to more than familiarity with the idea of data reuse or the skills needed to engage in reuse. "For us," one data services provider (Participant 9) explained, "it's awareness we need to solve with our customers." She continued: "That's of course key for us because if people are not aware of our data products, we are not making any money, so we have a natural interest to address this awareness problem through marketing and outreach." If, for some intermediaries, awareness was framed in terms of a particular product or service, then for others it was a broader question of "convincing people that reusing data is good for them" (Participant 3). Here, the intermediary's role in creating awareness can be seen to shift subtly from the provision of information to that of persuasion. This study participant went on to offer an account of the interplay between awareness and other influencing factors: "But if you convince [researchers] and you don't have the tools, then you will lose that awareness you have created." Awareness, on this account, is a necessary but not sufficient condition for data reuse to take place.

*Mindset* and *awareness* are individual-level constructs implying the existence of some process by which researchers come to see themselves as data reusers. One intermediary (Participant 17) narrated this process as a sort of epiphany, where researchers "suddenly realize that their knowledge and this other person's knowledge can make something new, and that can be exciting." Yet the active role that intermediaries saw for themselves in bringing about such realizations prompted us to look for other constructs beyond the individual level. One used by this intermediary (Participant 17) was *advocacy*, here in the context of prompting researchers to share their data in a particular way:

> So if someone says, oh, I have that data and you know, I can send it to you, I always say: please deposit it in the university repository, or at the very least somewhere like [the general-purpose repository] Zenodo, but make sure that you put it somewhere that it's

accessible, that you have metadata, that you make it findable and you give it a [persistent identifier] of some kind. Because I feel quite strongly about that, that we shouldn't be resorting to, you know, giving each other USB sticks or emailing data back and forth.

Through her use of this advocacy tactic, the participant sought to influence both the characteristics of a particular dataset and the future behavior of its creator. A different intermediary (Participant 8) noted that researchers "actually have to commit in the first place to doing that kind of work to make sure that then the data could be reused." While there is some slippage in these examples between data sharing and data reuse, our emphasis is on the role of the intermediary in influencing the researcher to do science differently. Our analysis suggests that there are dynamics of control and coercion at work here, perhaps most visible in a statement by the intermediary quoted above (Participant 17) expressing frustration that "for some reason, some researchers are blind to the benefits of doing this and doing it well." What comes into focus with statements like this is an institutional-level effort to remake the researcher in the image of data-intensive science, a process that—when coupled with the individual-level processes discussed above—is well captured by the inherently multilevel concept of subjectification. Framing the mechanism by which factors influencing data reuse are activated in this way does not mean that researchers who have formed a propensity to engage in data reuse do not genuinely wish to do so, nor does it mean that institutional efforts to bring about such a propensity are marked by pernicious intentions. Rather, and in keeping with the literature on subjectification [22], it affirms that scientific selves are co-produced by agentive individuals and by institutions that structure the very horizons of their agency.

Written responses to the question "Would you say that you see yourself as a data reuser?" from 8 of the participating researchers were used to validate our application of the construct of subjectification in coding the oral interviews. Of the 5 researchers who answered "Yes" to this question, 4 made statements that were associated with subjectification in their oral interviews (Participants 2, 11, 12, 23). Of the 3 researchers who answered "No" to this question, 2 did not make statements associated with subjectification in their oral interviews (Participants 1, 16) and the third made statements that were ambiguously associated with subjectification (Participant 4). By matching these responses across the two phases of data collection, we were able to check the internal validity of the construct.

The written responses provided further evidence of the relationship between subjectification and recurring reuse behavior. The majority of researchers who answered "Yes" to the question "Would you say that you see yourself as a data reuser?" reported that they had gone on to reuse data multiple times, while researchers who answered "No" to this question reported fewer instances of reuse. In fact, all three of the "No" respondents indicated that the reason they did not see themselves as data reusers was the infrequency of their reuse behavior; "I reused data once before," one explained (Interview 1), "but would not say that I do so on a regular basis."

Our analysis of written responses to a different question, "Can you briefly describe the situation or context that led you to form [the] belief [that data reuse could be a useful part of your research and/or teaching practice]?", highlighted the interaction between influencing factors and subjectification. For instance, we associated the response "beginning of my PhD research" (Participant 12) with the project-independent factor of community norms (see below). Norms around reuse encountered in the context of doctoral education may exert an influence on reuse behavior, but one that is mediated by a self-transforming process of enlistment into a field or discipline. Here, seeing oneself as a linguist (or member of some other research community) becomes coextensive with seeing oneself as a data reuser. Yet other researchers reported coming to see themselves as data reusers later in their careers. One (Participant 11)

described starting to collaborate with a colleague from a different discipline and realizing that it "totally made sense" to reuse a large panel dataset. We associated this response with the project-dependent factor of capabilities to reuse (see below), as this researcher explained that her ability to engage in reuse was initially bound up with her colleague's knowledge of the dataset and how to access it. The range of responses to this follow-up question points to the diverse settings in which subjectification can take place and the diverse influencing factors that can be activated by this mechanism.

## Factors influencing data reuse and their relationships

Many of the factors found to influence researcher data reuse in the existing scholarship were also identified as influencing factors in the analysis of our data. However, our analysis allowed us to draw out previously unexplored relationships between influencing factors that are *project-independent*, meaning that they affect whether a researcher will engage in data reuse regardless of considerations around a specific research project and/or dataset, and factors that are *project-dependent*, meaning that they affect whether a researcher will engage in data reuse in the context of a specific research project and/or dataset. Our analysis also suggests that while these influencing factors may have a direct effect on a single instance of data reuse, they have an indirect effect on recurring reuse. As discussed above, it appears to be the mechanism of subjectification that activates these influencing factors in a propensity to engage in reuse behavior.

## Project-independent factors

The three project-independent factors found to influence researcher data reuse spanned levels of analysis, from the individual-level attitudes of researchers to the norms of research communities and the rewards and requirements administered by the science system, which exert their influence at the institutional level.

**Researcher attitudes.** Attitudes on the part of researchers toward data reuse, understood as evaluative judgments about the practice, were found to be an individual-level factor influencing reuse behavior. Several researchers linked their reuse behavior to a positive attitude that associated reusing data with efficiency, although they took different perspectives on why efficiency was important. For instance, a health sciences researcher (Participant 2) explained that "you can save yourself a lot of legwork as a researcher if you ask: is there something like this in your organization? Do you have it already?" Here, efficiency reflects a rational calculation aimed at "saving time and maximizing output" (Participant 1). But others expressed a morally coded commitment to conserving resources at the level of the science system. For instance, an agricultural biotechnology researcher in a country with a low level of investment in scientific research (Participant 16) criticized the conduct of duplicative studies: "You cannot [justify] the resources to create the data in repetitive—it's just a waste of resources. I don't think so many scientists can afford to do that." Intermediaries tended to take the latter perspective, as with a data manager (Participant 3) who stated: "If you can reuse the data of others and make good use of it and extrapolate results that go beyond the initial use of that data, I think you're making a very efficient use of everybody else's time and work."

Other researchers linked their reuse behavior to a positive attitude that associated reusing data with a modern, cumulative approach to science in which "we can start to really ask very big questions" (Participant 13). Rather than concentrating scientific inquiry in places where specialized instruments are available, data reuse was also seen as a way of making research "a worldwide endeavor" (Participant 6). Yet even researchers who saw themselves as data reusers held some negative attitudes toward reuse. For instance, the health sciences researcher quoted

above acknowledged the significant effort required to reuse data, even when time is saved by not collecting it: "It's bad how long it still takes until a paper is published, for example; it is still a lot of work" (Participant 2). Another researcher (Participant 11) explained her shifting attitude toward data reuse with respect to the effort involved: "Before I thought to work with [reusing] quantitative data, I thought that this is a piece of cake. Because someone already did a lot of work for me. And now I only [need to] open the software and do a little bit of number crunching. Which is not the case at all." In these cases, subjectification appeared to offset the persistence of negative attitudes and set the stage for recurring reuse.

**Community norms.** Norms around data reuse within scientific communities, understood as the perceived incidence of and support for reuse on the part of a researcher's social referents, were found to be an institutional-level factor influencing reuse behavior. The self-reported influence of this factor on researcher behavior proved to vary in intriguing ways across our sample. Researchers in fields where data reuse is widely practiced tended not to point to community norms as a factor influencing their reuse behavior, instead emphasizing other, ostensibly technical factors such as data characteristics (see below). Yet researchers in fields where data reuse is not widely practiced did see community norms as blocking their reuse behavior. For instance, a sociology researcher (Participant 4) reflected that in his field "there is a prevailing culture of collecting one's own data for every research project. . .especially in the qualitative social sciences, often this possibility is not even considered, that one could actually fall back on existing data." This pattern suggests that negative community norms around data reuse may be more salient—but not necessarily more influential—than positive community norms, which become naturalized over the course of a researcher's socialization.

Intermediaries, on the other hand, clearly identified community norms as a factor influencing researchers' reuse behavior. As a university administrator (Participant 10) explained, "it's the tradition of certain scientific fields [rather than formal policies] which are dominating the practice." For instance, the director of an astronomical data center (Participant 6) described a widespread culture of reuse among astronomers dating back to the publication of printed catalogs. Yet her counterpart at an environmental sciences data center (Participant 7) acknowledged that "there is a subgroup where there's still this macho attitude about doing fieldwork, being out there getting your hands dirty [and collecting your own data]." Many intermediaries regarded community norms as one of the factors influencing reuse behavior that was most resistant to change. One (Participant 10) recounted a story of a meeting on open research practices across the disciplines where the question was posed: "Why can't the chemists be like physicists?" Dismissing the idea that community norms could simply be exported, this intermediary regarded exposure to role models within one's community as a trigger for subjectification, a means of bringing about not only a propensity for reuse by individual researchers but also broader cultural change.

**Rewards and requirements.** Rewards and requirements administered at the level of the science system, including research organizations, publishers, funders, and policymakers, were found to be an institutional-level factor influencing reuse behavior. On the whole, study participants saw system-level rewards as less influential than system-level requirements. Most researchers felt that conventional metrics like publications remained the primary avenue for career advancement, meaning that data reuse was only rewarded to the extent that it helped researchers to attain them. Here, again, field differences were apparent. A humanities researcher (Participant 23) noted that "I don't feel like I get rewarded much in terms of my career for creating the data, it's more the analysis of it. If I could find cases to reuse, I think I'd be quite happy." But a research funder (Participant 19), speaking about the field of paleontology, observed that "the field kind of falls into two camps: those people that do original analyses and those people that do these kinds of meta-analyses or syntheses. And the people that do the

original analyses just have no clue how much effort it takes to do synthesis. So yeah, there's that problem of just not really appreciating the work fully for the complexity of it, and not giving it enough credit basically."

Intermediaries were more likely to discuss changes in how research outputs are being evaluated as part of decisions around employment and resource allocation. One (Participant 7) observed that "people are starting to look at actually changing the tenure practices themselves to give broader credit," and pointed to the prospect of giving researchers credit "not just for being cited but for their data being reused more." Another system-level reward mentioned by a different intermediary (Participant 15) are the Research Parasite Awards, which were launched in 2017 to recognize outstanding scholarship carried out on the basis of data reuse. Yet it is striking that researchers themselves did not point to system-level rewards like these as factors influencing their reuse behavior, perhaps indicating their limited influence. However, one researcher (Participant 13) did describe a more fundamental process of reorientation toward a different set of rewards linked to data reuse and to open science more broadly:

> 99 percent of the time it's about the publication record. . .[but] I didn't like that anymore and struck out by myself. And now because of all these open scientific practices, which I've been doing for the last six or seven years, I've now defined a whole new career path for myself. I'm here at this place which I'd never even heard of a year ago, where they reward researchers and incentivize you. And their mission statement is open science, and they really value these sorts of things. [So] by not just practicing these things but also actively supporting them and advocating for these practices, it's helped to send my career off on a path which, five years ago, I would not have expected at all.

Although this researcher did not submit a follow-up response and so we cannot be sure that he saw himself as a data reuser, his account suggests that a process of subjectification activated the influence of an alternative set of system-level rewards on both his research practices and the direction of his scientific career.

In terms of system-level requirements, participants were divided about the influence of data sharing policies and mandates on reuse behavior. One intermediary (Participant 18) clearly laid out the case for seeing such requirements as an influencing factor: "I think that policies, like publisher policies, sometimes push other people to publish their data. Then that translates to the reuse of data." Yet the researchers in our study did not point to such requirements as factors influencing their own reuse behavior, even when the researchers were aware of their existence. Referring to journals with such policies in place, one researcher (Participant 2) offered a telling counterfactual: "That would be a source where I think, if I had the time, I would browse a lot and look for something [to reuse]." Intermediaries also tended to see legal requirements around intellectual property and data protection blocking reuse to a greater extent than researchers did, perhaps indicating their deeper familiarity with these requirements. Researchers held differing views about the influence of such requirements. One (Participant 4) contended that prospective reusers are "afraid of doing something wrong and then being confronted with sanctions," calling this "the biggest hurdle" for researchers. Yet others seemed to carry on what one intermediary (Participant 19) described as "a tradition of not caring" about the details of copyright and licensing. These researchers tended to focus on avoiding substantive harms to the data creator, whether by respecting an informal right of first publication (Participant 20) or by taking extra precautions with sensitive data (Participant 11), but they did not see legal requirements as such influencing their reuse behavior.

## Project-dependent factors

The four project-dependent factors found to influence researcher data reuse, including data characteristics, trust in the data, suitability for purpose, and capabilities to reuse, exerted their effects when researchers were considering a particular dataset for reuse. Yet these factors can be activated in the context of different research projects and/or datasets over time and can, therefore, influence recurring reuse behavior. In conceptual terms, factors of this type may be said to exert their influence at the project level of analysis, although this influence is exerted in interaction with both individual and institutional conditions.

**Data characteristics.** The objective characteristics of data being considered for reuse in the context of a particular research project were found to be one of the most important project-level factors influencing reuse of the data. Several participants discussed these characteristics with reference to the FAIR Guiding Principles for scientific data management and stewardship, i.e., findability, accessibility, interoperability, and reusability [53] (Participants 3, 6, 7, 12, 14, 21). While not all participants framed data characteristics in this way and one was actively critical of the FAIR approach (Participant 7), in this section we use the four high-level principles as a device for structuring our findings about data characteristics as a factor influencing reuse behavior. With respect to *findability*, one researcher (Participant 5) stated that "sometimes if the data is not. . .it's publicly available, but you don't know where you can find it. . .that will hinder the usage of this data." Conversely, another researcher (Participant 22) noted that "sometimes certain datasets are used more, although they are not the best ones, because they simply have better PR." Intermediaries described the range of strategies that they used to make the infrastructures they managed—and the data that these contain—findable, including rich metadata (Participant 6), persistent identifiers (Participant 10), and search engine optimization (Participant 14). "If you put the data in a repository and nobody knows it exists," one data manager (Participant 3) put it plainly, "nobody will reuse the data." Yet researchers reported that the proliferation of repositories could also make searching less efficient. "I would like to have only one way to find data," one (Participant 19) explained, "but at the moment that is not the case, and I don't see that happening soon."

*Accessibility*, in the context of the FAIR principles, does not necessarily refer to openness but rather to transparent processes for gaining authorization to retrieve data if controlled access is in place. Yet a number of researchers in our study reported the cost of access to data blocking reuse, with one (Participant 11) even indicating that the prospect of losing access to costly data resources was a deterrent from applying for other positions. Some intermediaries were experimenting with arrangements for promoting greater openness, as with one data center (Participant 3) that treated cost recovery for data deposits differently depending on whether the data were shared openly. Participants also discussed persistence of access to data, including earlier versions, as a factor influencing reuse; one researcher (Participant 12) described purchasing a dataset and downloading a local copy for future use due to worries about the long-term viability of the repository where it was stored.

A number of researchers noted that a dataset's "comprehensible structure" (Participant 1) or "standardized format" (Participant 4) could facilitate its reuse, and one environmental sciences researcher (Participant 22) described the challenge of harmonizing data collected in different countries as inputs to a research program aiming to "make global statements, that is, for the entire Earth." Overall, though, intermediaries tended to be more explicit than researchers about *interoperability* as a factor influencing reuse behavior. The director of an astronomical data center (Participant 6) spoke at length about the community-based process of defining and implementing standards that are, in her view, one of the main factors influencing reuse in her field. Yet the commitment of astronomy researchers to this process may also be an extension

of seeing themselves as data reusers, which prompts them not only to engage in reuse but also to treat the shared resource of their field's data with what Participant 6 described as "lots of care."

The *reusability* of a dataset was, unsurprisingly, seen by researchers and intermediaries alike as a factor influencing reuse behavior. Characteristics like file size can render a dataset less reusable, as the manager of an environmental sciences data center (Participant 7) explained; since researchers in this field "can't generally handle petabytes of data," enhancing reusability means dividing data up into "smaller spatial chunks, by continent, by region." Participants also emphasized the importance of what one intermediary (Participant 18) called the "chain of construction," whereby prospective reusers can trace how and under what conditions the data were generated. Our analysis revealed two different approaches to gathering this context. Researchers tended to value communicating directly with the data creator, in order to capture what one (Participant 20) described as "the experience that people had during the collection of the data. . .the things that go well or that don't go well during the collection phase." Knowledge of this nature tended to remain tacit, in this researcher's view: "Putting all your experience on paper takes too much of an effort." Yet intermediaries tended to favor the production of written documentation, or what one research aggregator (Participant 8) referred to as an "instruction manual." For these participants, reusability required making tacit knowledge explicit and alienable from its source, an approach that they sought to propagate as they encouraged researchers to understand themselves as data reusers. A similar impulse animated efforts to identify and capture any additional tools needed to support further analysis of the data. If, as one physical sciences researcher (Participant 24) suggested, "our data, by itself, is practically not understandable without the associated software," then reusability would have to extend beyond a characteristic ascribed to a given dataset to the reconstruction of an entire research environment.

**Trust in the data.**   While the objective characteristics of data being considered for reuse were found to be one project-level factor influencing whether reuse takes place, these characteristics were complemented by a more subjective assessment of whether or not the researcher could trust the data. This determination, which was shaped by both individual and institutional conditions, was found to be another project-level factor influencing reuse behavior. As one researcher (Participant 12) explained:

These [data characteristics] are maybe givens, right? Persistence, access, tagging, metadata. That's before I actually then look at the data and see if it makes sense, or if there's something weird there.

Researchers described a spectrum of quality assurance practices, ranging from a quick check for robustness to rerunning analyses to see whether reported results were reproducible. Local irregularities could be taken to index more general problems, as one researcher (Participant 23) explained: "In many cases, I can't be assured of that, from my own standpoint, why is this record there and that one isn't? Once I find a few of those issues, then I feel like I can't trust the data source as a whole." In the absence of trust, researchers might either move on to other data sources or seek to validate the data themselves, as another researcher (Participant 13) described undertaking:

What I had to do was, my goodness, a year, a year and a half of painstakingly going through every single published record I could find in my area of interest and making sure that the data were in there and that they were accurate and as complete as possible. That was critical because there was no way in hell I was going to just go into this database where I had no

idea what the quality was. Just pull the data out and analyze it and then try and publish? That's just reckless.

Because researchers saw data reuse in terms of staking their scientific reputation on someone else's work, they looked to intermediaries to mitigate this risk by vouching for the trustworthiness of data being considered for reuse. To this end, intermediaries at data centers and repositories outlined processes of internal review to ensure a baseline level of quality. The editor of a data journal (Participant 14) described a more stringent process by which independent experts write formal reviews of submitted datasets, which are signed and published alongside the data. A program director at a scientific publisher (Participant 21) observed that traditional journals reviewing the data behind the articles they publish are addressing issues of trust and reliability "in a really serious way," although he cautioned that at present "those people are not the majority."

While these practices of quality assurance could enhance trust and thereby influence reuse, researchers also reported that their trust in a given dataset could stem from its association with institutions that they regarded as trustworthy. "If I go to my dataset and see [a leading European research organization]," one researcher (Participant 12) reflected, "it comes with a stamp of approval, right? It's not John the butcher or Joe the baker." Another researcher (Participant 11) wished for a search tool that would give prospective reusers "the top recommendations of different types of data sources or data depositories which we trust." Earlier in the interview, this researcher had described a disappointing experience using a commercial database, which was said to provide "top-quality data. . .like crème de la crème," but in fact delivered spotty data that required extensive checking. Thus, even as intermediaries sought to position their organization as a "trust provider" (Participant 9), researchers found themselves needing to check these assertions against their own expert judgment. Yet this judgment could be marked by its own blind spots, as a data manager at a South African repository (Participant 18) clarified: "We are often faced with quite underdefined and, I think, biased ideas from the global North toward data coming from Africa that, somehow, by the very fact that it's this low-resource environment, [the data] must be of questionable quality." In this light, even researchers who see themselves as data reusers may not follow through with reuse in the absence of a critical evaluation of the trust signals on which they rely.

**Suitability for purpose.** Even when the characteristics of a given dataset facilitate its reuse and a prospective reuser has judged the data to be trustworthy, the data's suitability for the researcher's intended purpose was found to be another project-level factor influencing reuse behavior. That is, a dataset judged to be suitable for one purpose might be seen as entirely unsuitable for another. The purpose for which researchers were found to be the most selective was new research undertaken with a well-defined research question. A health sciences researcher (Participant 2) explained that "we first think about what we want to know, and only then describe the study population, for example. Just screening a data set and seeing, could this be significant or that be significant, that's not our way." Similarly, a humanities researcher (Participant 23) reflected that "humanists have often a question or an interest in mind, and if the data doesn't speak directly to that, we won't use it—even if it's close." Yet researchers also described other scenarios where determining the data's suitability was a more iterative, exploratory process. For one business researcher (Participant 11), "there might be some sort of intuition and then I need to have a look in the data, [to see] if there is any particular support." Another explained that "if you do it according to the textbook, then you develop a hypothesis model beforehand. . .and on the basis of the data you test the hypotheses and verify or falsify them. But in practice it doesn't happen like that. In practice you start from the dataset, you see what's in there. . .and what you can make out of it" (Participant 1).

Another purpose for reusing data discussed by some researchers was teaching. "What I may do in teaching with data," one researcher (Participant 23) explained, "is to bring in some sample datasets that students can use to try out different software, or I may point them to different sources of data, so that they can research and find a data set that interests them." This participant noted that, because his students are "interested in their own location," local datasets have proven to be particularly suitable for reuse in the classroom, as well as data that are accessible without needing to contact their creator; students, he explained, are "often on a much shorter deadline with a project, and so there's often not time to have that back and forth." Another researcher (Participant 4) made his own focus group data available for students to reuse. The suitability of this dataset derived, in part, from how well the researcher knew it, allowing him to direct students toward "what might be interesting insights, latent meaning structures that are interesting in certain passages, in order to give [them], so to say, a sense of achievement."

A few researchers indicated that they had sought out existing data for the explicit purpose of engaging in data reuse. One medical researcher who now sees himself as a data reuser (Participant 20) noted that he started working with a particular panel study "to see how easy it is to reuse open data. So, in fact it was the main goal of this activity, to get experience." Similarly, a business researcher (Participant 5) elected to participate in an initiative where a pharmaceutical company made a proprietary dataset openly available with the goal to "see how anyone can actually experiment with these data to come up with something new." And while intermediaries generally saw their role in terms of facilitating reuse for whatever purpose researchers defined, some also took steps to promote data reuse for particular purposes in line with their organization's mission. For instance, an archivist at an environmental sciences data center (Participant 7) noted that "we're looking for data that helps users work with remote sensing data but integrate it with, say, different kinds of socioeconomic data in order to facilitate certain kinds of research or applications." Here, actively fostering integrative or interdisciplinary reuse was one way for the data center's sponsoring to amplify the relevance of its research for policy issues. More broadly, examples like this show that the purpose for which existing data are or are not judged suitable can be co-created by researchers and intermediaries, just as with the process of subjectification itself.

**Capabilities to reuse.** When the objective characteristics of a dataset being considered for reuse are aligned with subjective assessments of the data as trustworthy and suitable for the researcher's purpose, one other project-level factor influencing reuse behavior was found to be capabilities to reuse the dataset in question. Capabilities, here, refer not only to the individual researcher's abilities but also to institutional arrangements of labor and technology that structure capacities for action. Our analysis revealed three sets of capabilities that influenced researchers' data reuse; these seem to exist in a complementary relationship, such that strength in one or two may offset a relative lack of strength in another.

Researcher *skills and training* were seen by many study participants as a factor influencing reuse behavior. These individual-level capabilities could be acquired in various ways: for instance, one researcher (Participant 11) discussed how collaborating with a more experienced colleague helped her to access and reuse a restricted dataset more easily. Others engaged in more formal training opportunities, as with the researcher (Participant 13) who described a five-week workshop on computational analysis that he completed during his PhD as "transformational" in terms of his ability to reuse data. He went on to explain:

When I started my PhD, I was just expected to learn these things myself. It's like: "Here's this database, here's what I want you to do, now go ahead and do it." And I was like: "Shit, man, I'm a geologist. I never learned how to program or to write scripts."

Another researcher (Participant 2) admitted that she would not know how to handle a concern identified over the course of reuse about the findings of the data creators: "What do I do if it comes out that their calculations are garbage? Am I ethically obliged to write to the journal then?" Some intermediaries saw skills and training as the clear answer to uncertainties like these, which could hinder data reuse. Thus, one data curator (Participant 17) asserted that what drives reuse is "a learnable set of knowledge and competences that can be done well or badly." Crucially, for these intermediaries, settings for the transmission of knowledge about reuse could also serve as settings for subjectification.

Other intermediaries emphasized the availability of *support services* as institutional-level capabilities that could facilitate reuse behavior. On this view, the researcher's skills and abilities can be augmented by the specialized labor of dedicated data professionals, who may be the ones to "get their hands dirty and wrangle data. . .for often busy principal investigators" (Participant 18). Intermediaries taking this view maintained that the provision of support services deserved to be seen as scientific work in its own right, with one (Participant 10) forecasting that "the moment the data expert's name is number one or two on a scientific publication, then we are there, because then recognition is given to this important work of data integration." Yet it is notable that only one of the researchers interviewed for this study (Participant 12) discussed making use of support services provided by a data steward, and this was in the context of sharing data rather than reusing it. The fact that most researchers did not point to this set of capabilities as a factor influencing their data reuse may suggest that such support services are still nascent and unevenly distributed, or that researchers are not receptive to them.

A third set of capabilities described by some participants as a factor influencing reuse behavior involved the development of *easily usable tools* for working with data as part of a next generation of digital infrastructure. One data manager (Participant 3) explicitly contrasted this approach with an emphasis on the cultivation of researcher capabilities, noting: "If the database is sufficiently well-done, you should not need training." In this vein, several intermediaries mentioned data visualizations as a tool that could help researchers to more quickly assess the value of delving deeply into a given dataset and, potentially, reusing it. Others touched on more convenient delivery mechanisms, from APIs (Participant 9) to a fully cloud-based environment where data, tools, and services could be collocated (Participant 7). However, both researchers and intermediaries voiced concerns about the growing dominance of commercial entities and products in this space, given the fact that "if you're not a subscriber or if, for example, [one service] goes away, then most people with projects in it get shut down" (Participant 18). Here, recurring reuse is understood to be at risk, even when researchers taking advantage of these capabilities see themselves as data reusers.

## Discussion

Whenever a scientific researcher embarks on an empirical study, one of the decisions they in principle face is whether to collect new data or reuse data that exists already. Previous research has identified a number of factors that influence this decision. Our study finds that such influencing factors (which we characterize in terms of their relationship to a specific research project) have an indirect effect on recurring data reuse as mediated by subjectification. For the purposes of this study, we define subjectification as a multilevel process whereby researchers come to see themselves as data reusers and intermediaries seek to bring about this self-understanding on the part of researchers. Our findings suggest that influencing factors may have a direct effect on a single instance of data reuse, but point to subjectification as the mechanism by which influencing factors are activated in a propensity to engage in data reuse. For, if the scientific and societal benefits associated with data sharing are to be broadly realized, then it is

the ongoing inclination to engage in data reuse—not invariably, but as a matter of normal science—that demands to be understood.

A key advantage of the model of recurring data reuse behavior derived from our findings (see Fig 1) is the distinction it draws between factor type and level of analysis. Each of the seven influencing factors is assigned to one of two factor types: project-independent or project-dependent. This analytical move extends previous research [17] that did not propose to structure or characterize the factors influencing data reuse. Each of the seven influencing factors is also assigned to one of three levels of analysis. Project-independent factors exert their influence at either the individual (i.e., researcher attitudes) or institutional (i.e., community norms; rewards and requirements) level, while project-dependent factors may be said to exert their influence at the project level. For the sake of clarity, each factor is associated with only one level of analysis, although project-level factors (e.g., trust in the data; capabilities to reuse) were found to exert their influence in interaction with both individual and institutional conditions. But our findings are able to explain how all seven factors exert an influence on recurring data reuse behavior across levels of analysis, because the activating mechanism of subjectification is an inherently multilevel process by which self-understanding is produced out of inputs at multiple levels of analysis. This conceptual move extends previous research [39] showing cross-level effects of factors influencing researcher data reuse through statistical analysis alone. By introducing a concept that insists on the indissociability of multiple levels of analysis in understanding how researchers come to see themselves as data reusers, our study reveals the mechanism by which individual, project, and institutional-level factors are joined up in a whole, complex person, subject to both agency and constraint.

One limitation of our study is the imprecision of subjectification as a construct, couched as it often is in plain language terms like self-understanding. Future research could include a process of scale development resulting in a more precise operationalization of the construct, as has been undertaken for the related concept of self-identity [54]. This, in turn, would address another limitation of the present study: its small sample size. A larger quantitative study, designed around survey items carefully constructed to exclude neighboring concepts, would make it possible to test the effects proposed in our model and to advance causal claims. A longitudinal design would lend itself to measuring reuse recurrence and determining whether the activating role of subjectification stays constant, weakens, or strengthens over time. In her discussion of methodological designs for studying subjectification, Sigl [23] endorses this longer-term approach, although she suggests allowing for enough flexibility to explore environmental inputs that may not have been identified at the outset.

Even if the links between influencing factors, subjectification, and recurring reuse behavior can be decisively established, the practical challenge remains how to apply this knowledge to the organization of scientific research. Currently, institutional efforts to promote researcher data reuse tend to revolve around building infrastructures like the European Open Science Cloud [55] and enriching the data that they contain [56]. But our findings suggest that technical solutions alone may not lead to a widespread change in existing practices. Following our analysis, institutions such as universities and learned societies may wish to create purpose-built settings for subjectification in which researchers can form a propensity to engage in data reuse. What might these look like? Here, too, more research is needed, but one possible model can be found in recent calls for building the capacity of emergent data communities [57]. When a loosely connected group of researchers shares a stake in working with a particular type of data, then institutional support has the potential to intensify experimentation and exchange. Another model is to create safe learning spaces for experimenting with and applying a range of open and collaborative scientific practices including data reuse, as has been demonstrated at the Lab for Open Innovation in Science at the Einstein Center for Neurosciences

Berlin. Both of these models have potential for creating conditions favorable to subjectification. Yet such conditions cannot be manufactured out of thin air; indeed, they may require an openness to critical deliberation about data reuse and the forms of power that are being enlisted to promote its adoption. Settings for subjectification, if they are to be successful, cannot only seek to transform researcher behavior; they must also empower researchers to interrogate the uses of reuse in a changing science system.

## Supporting information

**S1 File. Interview guides for semi-structured oral interviews.**
(PDF)

**S2 File. Written follow-up questions for researcher participants.**
(PDF)

**S3 File. Final coding table.**
(PDF)

## Acknowledgments

First and foremost, we want to thank our study participants, who generously gave of their time to share the experiences and perspectives that form the basis of the study's insights. Katie Hyslop validated the oral interview coding, Robin Brehm created the online form for collecting the written follow-up responses, Susanne Beck contributed to the development of the model, and Lisa Sigl offered feedback on our use of the concept of subjectification. Lisa Hönegger provided guidance on archiving and sharing our data, and Roja Ratzinger deidentified the oral interview transcripts. Earlier versions of this article were presented at the Leibniz Research Alliance Open Science Conference and the LBG Open Innovation in Science Center Brown Bag Seminar series.

## Author Contributions

**Conceptualization:** Marcel LaFlamme, Marion Poetz.

**Data curation:** Marcel LaFlamme, Daniel Spichtinger.

**Formal analysis:** Marcel LaFlamme, Daniel Spichtinger.

**Investigation:** Daniel Spichtinger.

**Methodology:** Marion Poetz.

**Project administration:** Daniel Spichtinger.

**Writing – original draft:** Marcel LaFlamme.

**Writing – review & editing:** Marion Poetz, Daniel Spichtinger.

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
