## [Decision Letter · Decision Letter 0]

19 Aug 2021

PONE-D-21-21899

Seeing oneself as a data reuser: How subjectification activates the drivers of data reuse in science

PLOS ONE

Dear Dr. LaFlamme,

Thank you for submitting your manuscript to PLOS ONE. After careful consideration, we feel that it has merit but does not fully meet PLOS ONE’s publication criteria as it currently stands. Therefore, we invite you to submit a revised version of the manuscript that addresses the points raised during the review process.

Your manuscript was reviewed by three accomplished scientists with expertise in data reuse. I also reviewed the paper, and I concur with the vast majority of all comments, critiques, and suggestions raised by the reviewers. This is a significant manuscript that presents much-needed findings in an area seldom studied. I strongly agree with Reviewer 1’s concern about the lack of support of some statements made in the Discussion section. Also, in this section, the specific insights gained from the study findings, and their relationship with previous research on data reuse, should be discussed in more detail.

We look forward to receiving your revised manuscript.

Kind regards,

Sergi Fàbregues

Academic Editor

PLOS ONE

Journal Requirements:

2. Please include additional information regarding the interview guide used in the study and ensure that you have provided sufficient details that others could replicate the analyses. For instance, if you developed an interview guide as part of this study and it is not under a copyright more restrictive than CC-BY, please include a copy, in both the original language and English, as Supporting Information.

Reviewers' comments:

Reviewer's Responses to Questions

**Comments to the Author**

1. Is the manuscript technically sound, and do the data support the conclusions?

Reviewer #1: No

Reviewer #2: Yes

Reviewer #3: Yes

2. Has the statistical analysis been performed appropriately and rigorously? 

Reviewer #1: N/A

Reviewer #2: Yes

Reviewer #3: N/A

3. Have the authors made all data underlying the findings in their manuscript fully available?

Reviewer #1: Yes

Reviewer #2: Yes

Reviewer #3: No

4. Is the manuscript presented in an intelligible fashion and written in standard English?

Reviewer #1: Yes

Reviewer #2: Yes

Reviewer #3: Yes

5. Review Comments to the Author

Reviewer #1: Overall, this manuscript describes a study that is well done and pertinent to the readers of PLOS ONE. The approach is thoughtful and timely. Giving that reuse is one of the main rationales for mandating data sharing policies, more careful investigations of data reuse behavior is certainly necessary. As the authors note, and a few others have before them, there is a lack of literature in this area. I was happy to see the article, and as I read, I found myself thinking of people I would want to send this work to when it comes out. However, I still see some major issues that should be addressed before publication.

There are several strengths:

Approach -

I suppose it could be somewhat controversial, but I particularly liked that there was purposefully no attempt to ground the study in an existing framework or set of constructs. In the literature, there seems to have been a priori assumptions that reuse behavior could fit into an existing models, and I agree with the authors that this is a limitation in the current lit. It also gets past the fuzziness around definitions of data and data reuse to begin with and lets the interviewees define their behavior on their own.

Methodology -

As described, the methods are appropriate and sound. That said, I have some issues with the analyses, or at the very least, the presentation (see below).

Intro and review of the current literature to date -

The coverage is well done, as is the gap analysis and the rationale for the approach used here. A few times it seems a little too editorial (e.g. referring to Zuiderwijk et al review as “somewhat static” - FYI, I am not associated with that work nor do I know any of the authors).

However, there are also weaknesses:

Reviewer access to interview guide and data -

Without access to the interview guide or reference to the specific interview prompts in the text, it was disorienting to read through the results without context. As a tiny example, I was surprised to see two different interviewees both used the specific word "tradition," and I wondered if that specific word was used in the prompt. Maybe not - or perhaps this was a translation? Either way, this just an example of how, as a reader, I had no way of knowing what the interview subjects were reacting to exactly, and it left questions unanswered as I read.

Comparison to other results -

I was somewhat disappointed in the results. I hoped by using an approach to "elicit the perspectives of study participants in words of their own choosing" - new data would be collected. However, these data seem very similar to that collected in other studies. I suspect (but cannot tell) that this is largely because of a similarity in interview guides between this study and others? In fact, I caught myself thinking that with these results, the authors could have reused others’ interview data themselves. This is not something to “fix” - it is a reality, and the nice thing is that these results are consistent with other studies then. The approach to analysis is different, which is valuable, so I don't think this is necessarily a weakness in the study overall, but I think the similarity to other data is worth briefly noting in the discussion.

Conclusions -

By far, my biggest concern is a lack of support for these conclusions:

"we find that project-independent factors like researcher attitudes and community norms exert a lasting influence on reuse behavior only when researchers are subjectified (i.e., come to see themselves) as data reusers"

"that project-dependent factors like data characteristics and suitability for purpose come to facilitate a pattern of reuse only when subjectification takes place"

Specifically - the claim that reuse behavior happens *only when* researchers are subjectified/subjectification takes place and researchers see themselves as "data reusers."

The results indicate that half of participants expressed sentiments consistent with subjectification, and from the quotes provided, this seems valid. However, what evidence of timing was found? Is it not possible that behavior and subjectification happens concurrently? Or reuse behavior happens even without researchers being explicitly conscience of being a "data reuser" (and as such, thinking of themselves as one) until afterwards?

For the half whose interviews did not contain any data that suggested subjectification, did any specifically see themselves as *not* data reusers? Certainly, lack of evidence is not evidence of absence, but it would be worth mentioning, and I can't quite tell even if the text is attempting to, e.g. Interview 4 is a slippery example - are they or are they not considered a data reuser by themselves and for the purposes of this study? That it's context-specific is clear, which means it could be used to argue for or against your claim.

Ultimately, these results have not convinced me that subjectification is a prerequisite to *reuse behavior.* I am certainly intrigued by the concept of subjectification as it pertains to data reuse - I have no bias against this claim. Moreover, I think there is something here, just that it is possibly over-stated or not presented clearly enough. Given the emphasis on subjectification as the conclusion I would like to see the support flushed out much more, so much so that I would categorize it as a "major" revision.

A few other minor points -

- I believe ref 8 has the wrong year - 2006 not 2004

- RE: "Many interviewees discussed ... with reference to the FAIR Guiding Principles" - how many and which? Same for the next sentence "While not all ... a few were actively critical ..." The authors were otherwise very good about indicated which interviewee was being referenced throughout.

- Limitations should be addressed

Reviewer #2: This study conducts semi-structured interviews and multiple rounds of coding with 24 researchers and mediators, finds that project-independent factors and project-dependent factors, and finds that they were influenced by the stimuli of subjectification, constituting a model of data reuse behavior. However, the current version of the manuscript needs some improvement before being considered for publication, and the following three aspects could be considered.

1. It is nice to see that the authors introduce a theoretical discussion based on a critical analysis of current research on the relevant factors affecting data reuse, and thus introduces the concept of subjectification in the section "Drivers of data reuse in science". I suggest that it would be better to divide all the review or analysis into two subsections, the elemental discussion and the theoretical discussion, and name them under two subheadings to present the content more clearly.

2. The article focuses on multilevel drivers of data reuse and subjectification as a key mechanism by which these influencing factors are activated. A behavioral model of data reuse is developed in the Results section. However, there is a lack of explicit analysis or description of the multilevel nature of data reuse drivers, as well as a lack of clarity on how to conclude that subjectification is a key mechanism for activating these influences and how subjectification activates these factors. It is recommended that a description of Figure 1 or be added from both two points.

3. Both the first paragraphs of Project-dependent factors and Project-independent factors claim to discuss these factors and their interactions (p.12, line 265 and p.17, line 390), but in the subsequent subsections, no discussion of interactions of those factors is seen. The discussion of their interactions could better reveal the multilevel characteristic of the data reuse drivers.

Reviewer #3: The process of coding is not clearly stated, where are the results of applying Atlas.ti software?

Lines from 62 to 80 from Introduction seem part of the results, and from 81-84 part of conclusions

I have missed some tables to tabulate the qualitative analysis, it would easier to follow the results by tabulating factors and comments by interview and type of participant. I recommend to do this.

Data are not available, according to authors, data will be sahred only if the paper is acepted, however if the paper is accepted it has not sense to review the data. This seems cotradictory with the subject of the paper

6. PLOS authors have the option to publish the peer review history of their article (what does this mean?). If published, this will include your full peer review and any attached files.

Reviewer #1: **Yes: **Heidi Imker

Reviewer #2: No

Reviewer #3: No

---

## [Author Response · Author response to Decision Letter 0]

8 Mar 2022

We would like to thank the editor and the three reviewers for their helpful comments and suggestions. Please see our point-by-point responses below.

Editor Comments

This is a significant manuscript that presents much-needed findings in an area seldom studied. I strongly agree with Reviewer 1’s concern about the lack of support of some statements made in the Discussion section. Also, in this section, the specific insights gained from the study findings, and their relationship with previous research on data reuse, should be discussed in more detail.

• Thank you for these encouraging comments; we hope that our work can further advance the conversation about this understudied scientific practice.

• Reviewer 1’s comments prompted us to clarify the boundary conditions of our central claim and to collect additional data to support it, as we discuss below. We have revised the Discussion section to reflect the intended scope of our claims, which we believe are supported by our analysis of the written follow-up responses we collected.

• On lines 769–773, we also drew a clearer link between the insights of our study (i.e., the importance of activating mechanisms that draw together influencing factors at different level of analysis) and the value of reexamining previous research in that light.

Reviewer #1 Comments

Overall, this manuscript describes a study that is well done and pertinent to the readers of PLOS ONE. The approach is thoughtful and timely. Giving that reuse is one of the main rationales for mandating data sharing policies, more careful investigations of data reuse behavior is certainly necessary. As the authors note, and a few others have before them, there is a lack of literature in this area. I was happy to see the article, and as I read, I found myself thinking of people I would want to send this work to when it comes out.

• Thank you for these encouraging comments; we hope that our work can further advance the conversation about this understudied scientific practice. 

I suppose it could be somewhat controversial, but I particularly liked that there was purposefully no attempt to ground the study in an existing framework or set of constructs. In the literature, there seems to have been a priori assumptions that reuse behavior could fit into an existing model, and I agree with the authors that this is a limitation in the current lit. It also gets past the fuzziness around definitions of data and data reuse to begin with and lets the interviewees define their behavior on their own.

• We are glad that the reviewer sees the value of our inductive approach, as a complement to previous studies that started from a predefined theoretical framework. Thank you for pointing out the connection between this approach and the definitional issues mentioned in the Introduction; on line 199, we have clarified that we used a self-report of previous data reuse as an inclusion criterion for the study.

As described, the methods are appropriate and sound.

• We are glad that the reviewer sees the value of our interview-based approach, given the aims of the study.

Intro and review of the current literature to date: the coverage is well done, as is the gap analysis and the rationale for the approach used here. A few times it seems a little too editorial (e.g. referring to Zuiderwijk et al review as “somewhat static” - FYI, I am not associated with that work nor do I know any of the authors).

• Thank you for pointing this out; we have made a number of text changes throughout this section to present previous research in this area in a more neutral fashion, including removing this characterization of the Zuiderwijk et al. review.

Without access to the interview guide or reference to the specific interview prompts in the text, it was disorienting to read through the results without context. As a tiny example, I was surprised to see two different interviewees both used the specific word "tradition," and I wondered if that specific word was used in the prompt. Maybe not - or perhaps this was a translation? Either way, this just an example of how, as a reader, I had no way of knowing what the interview subjects were reacting to exactly, and it left questions unanswered as I read.

• We have provided the interview guides as supporting information, although it is worth noting that the interviews were semi-structured by design and therefore the sequence and wording of the questions was not completely consistent across the interviews.

• With respect to the specific example given, both Interview 10 and Interview 19 (from which we present quotations that include the word “tradition”) were conducted in English. In these interviews, the word “tradition” was independently supplied by the two study participants; the interviewer did not use the word in either case.

I was somewhat disappointed in the results. I hoped by using an approach to "elicit the perspectives of study participants in words of their own choosing" - new data would be collected. However, these data seem very similar to that collected in other studies. I suspect (but cannot tell) that this is largely because of a similarity in interview guides between this study and others? In fact, I caught myself thinking that with these results, the authors could have reused others’ interview data themselves. This is not something to “fix” - it is a reality, and the nice thing is that these results are consistent with other studies then. The approach to analysis is different, which is valuable, so I don't think this is necessarily a weakness in the study overall, but I think the similarity to other data is worth briefly noting in the discussion.

• We agree that the factors influencing researcher data reuse identified in our study are similar to those identified in previous studies.

• Our interview guide included a series of open-ended questions, which were accompanied by a bank of potential responses that the interviewer could check the interviewee’s response against and, time permitting, ask follow-up questions about. While the unprompted portion of the interviewee’s answer generally yielded the richest data, our use of these probes (which were identified through a review of the existing literature) likely did introduce some bias toward previously identified factors. We have added a discussion of this aspect of our study design on lines 224–226.

• However, we did not see it as important to shield our interviewees from these potential responses, precisely because of the maturity of the evidence base from which they were drawn. We regard the influencing factors themselves as settled science, and accordingly we have added language to the Introduction and Discussion sections clarifying that our findings with respect to influencing factors are largely confirmatory (although we do draw out cross-level relationships between factors that are project-independent and factors that are project-dependent, which to our knowledge have not been explored elsewhere).

• As discussed below, we regard our findings about subjectification as a mediator of the direct effects of these influencing factors as the study’s core contribution. Our primary goal was not to find new influencing factors, but to explore the mechanisms by which known factors are activated in the context of recurring reuse. We have sought to clarify this in the Introduction and Discussion sections, but also in the way that we restructured the Results section. 

• The point about reusing existing interview data is an intriguing one that has given us cause for reflection. At the time when the study was conceived, we were not aware of any openly accessible collections of interviews from a comparable range of participants, and it is unclear whether it would have been effective to request unreleased data from the authors of previous studies given the spotty availability of data upon request. However, in the spirit of reflexivity, we as researchers were also less knowledgeable about data reuse and its benefits at the outset of this study than we are today. Knowing what we know now, it is likely that we would at least evaluate the possibility of reusing existing data more carefully. We do see potential in reanalyzing previous research to validate our findings around subjectification and have noted this as one area for future research in the Discussion section.

By far, my biggest concern is a lack of support for these conclusions:

"we find that project-independent factors like researcher attitudes and community norms exert a lasting influence on reuse behavior only when researchers are subjectified (i.e., come to see themselves) as data reusers"

"that project-dependent factors like data characteristics and suitability for purpose come to facilitate a pattern of reuse only when subjectification takes place"

Specifically - the claim that reuse behavior happens *only when* researchers are subjectified/subjectification takes place and researchers see themselves as "data reusers."

The results indicate that half of participants expressed sentiments consistent with subjectification, and from the quotes provided, this seems valid. However, what evidence of timing was found? Is it not possible that behavior and subjectification happens concurrently? Or reuse behavior happens even without researchers being explicitly conscience of being a "data reuser" (and as such, thinking of themselves as one) until afterwards?

• Thank you for the insightful and constructive criticism on this point, which has prompted us to clarify the boundary conditions of our central claim. We have no doubt that reuse behavior takes place without researchers seeing themselves as data reusers; an intuitive example from our interviews, which we have added on lines 262-263, would be a PhD student who reuses data on a onetime basis as part of a required course assignment. In this context, reuse behavior can likely be explained most parsimoniously by the direct effect of rewards and requirements associated with academic coursework.

• Moreover, we agree that reuse behavior and subjectification can happen concurrently; this is reflected in our model (Figure 1) by the feedback loop connecting reuse behavior and subjectification.

• However, our central claim (which you rightly pushed us to formulate more clearly) is that subjectification has a mediating effect (i.e., the “only when” condition) on recurring reuse behavior, or what we describe as a propensity to engage in data reuse. While our oral interview data suggested this connection between subjectification and recurring reuse, we collected additional data to validate this mechanism. We invited the 12 researchers in our sample to provide written follow-up responses on whether and why they saw themselves as data reusers, how they formed this belief, and how many times they have engaged in data reuse after doing so; see lines 236-241 for details.

• As we discuss in lines 351-358, these written responses provided further evidence of the relationship between subjectification and recurring reuse behavior. That is, interviewees who saw themselves as data reusers were more likely to report reusing data on multiple occasions than interviewees who did not see themselves as data reusers. Moreover, interviewees who did not see themselves as data reusers explained that they did not see themselves in this way on the basis of their infrequent reuse.

For the half whose interviews did not contain any data that suggested subjectification, did any specifically see themselves as *not* data reusers? Certainly, lack of evidence is not evidence of absence, but it would be worth mentioning, and I can't quite tell even if the text is attempting to, e.g. Interview 4 is a slippery example - are they or are they not considered a data reuser by themselves and for the purposes of this study? That it's context-specific is clear, which means it could be used to argue for or against your claim.

• Thank you for pushing us to be more precise in our use of evidence here. As part of the written follow-up responses that we invited the researchers in our sample to provide, we specifically asked whether the participant saw themselves as a data reuser. Of the 8 researchers who responded, 5 indicated that they did see themselves as reusers and 3 indicated that they did not (including the participant associated with Interview 4; his quoted remarks on lines 290-291 raise the issue of subjectification but, as you rightly point out, do not make it clear that he saw himself as a reuser).

Ultimately, these results have not convinced me that subjectification is a prerequisite to *reuse behavior.* I am certainly intrigued by the concept of subjectification as it pertains to data reuse - I have no bias against this claim. Moreover, I think there is something here, just that it is possibly over-stated or not presented clearly enough. Given the emphasis on subjectification as the conclusion I would like to see the support flushed out much more, so much so that I would categorize it as a "major" revision.

Thank you for your willingness to engage with the concept work that we are doing. We hope that, by clarifying the boundary conditions of our central claim and by collecting additional data to support it, we have persuasively linked subjectification to recurring reuse behavior (while acknowledging that onetime reuse may well be explained by existing models that posit the direct effects of influencing factors. Of course, our study can only provide exploratory insight into this mechanism; we hope to inspire further research that operationalizes subjectification more precisely, as we indicate in the Discussion section.

A few other minor points -

- I believe ref 8 has the wrong year - 2006 not 2004

Thank you for pointing this out; we have corrected the publication year at line 831.

- RE: "Many interviewees discussed ... with reference to the FAIR Guiding Principles" - how many and which? Same for the next sentence "While not all ... a few were actively critical ..." The authors were otherwise very good about indicated which interviewee was being referenced throughout.

As requested, we have provided interview numbers at lines 526 and 528.

Limitations should be addressed.

• We have added a discussion of the study’s limitations, including our small sample size and the imprecision of the subjectification construct, to the Discussion section.

Reviewer #2 Comments

It is nice to see that the authors introduce a theoretical discussion based on a critical analysis of current research on the relevant factors affecting data reuse, and thus introduces the concept of subjectification in the section "Drivers of data reuse in science".

• Thank you for these encouraging comments; we hope that our work can help to focus the conversation around the role of theory in research on data reuse.

I suggest that it would be better to divide all the review or analysis into two subsections, the elemental discussion and the theoretical discussion, and name them under two subheadings to present the content more clearly.

• Thank you for this suggestion, which we have implemented.

There is a lack of explicit analysis or description of the multilevel nature of data reuse drivers, as well as a lack of clarity on how to conclude that subjectification is a key mechanism for activating these influences and how subjectification activates these factors. It is recommended that a description of Figure 1 or be added from both two points.

• In the Results section, we have added language clarifying whether identified factors were regarded as an individual- or institutional-level factor. Two of the factors, data characteristics and capabilities to reuse, appear to operate across levels of analysis.

• On lines 335–336, we have added an explanation of subjectification as an inherently multilevel concept, whereby researchers come to see themselves as data reusers and intermediaries seek to bring about this self-understanding on the part of researchers.

• As we discuss on lines 359–375, our analysis of written follow-up responses to the question “Can you briefly describe the situation or context that led you to form [the] belief [that data reuse could be a useful part of your research and/or teaching practice]?” highlighted the interaction between subjectification and influencing factors. By deductively applying factor codes to these responses, we were able to show that both project-independent and project-dependent factors can be linked to settings for subjectification and, in turn, to the formation of a propensity for reuse.

Both the first paragraphs of Project-dependent factors and Project-independent factors claim to discuss these factors and their interactions (p.12, line 265 and p.17, line 390), but in the subsequent subsections, no discussion of interactions of those factors is seen. The discussion of their interactions could better reveal the multilevel characteristic of the data reuse drivers.

• Thank you for pointing this out. Our use of the word “interaction” in this context was misleading, since we do not advance specific arguments about the co-occurrence of certain factors (i.e., are researchers who report being influenced by factor X more or less likely to be influenced by factor Y?). The primary goal of the study was not to discover factor interactions in this sense, but to explore the mechanisms by which influencing factors are activated.

• However, our analysis did point to the distinction between project-independent and project-dependent factors, in terms of the conditions under which they exert their influence. Notably, both groups of factors include factors that operate at the individual and institutional levels of analysis, reinforcing our paper’s broader point about the indissociability of these levels. We hope that characterizing these groups of factors in terms of “cross-level relationships” rather than “interactions” clarifies the kind of claim that we are making about them.

Reviewer #3 Comments

The process of coding is not clearly stated, where are the results of applying Atlas.ti software?

• As we discuss on lines 251–252, we used Atlas.ti to iteratively code and recode the interviews, which resulted in the final codebook that corresponds to the elements of Figure 1 and the structure of the Results section.

• Given our interpretivist approach to this study, we did not make use of the various pattern analysis or visualization features of Atlas.ti, although we recognize that these features could be useful in a different context.

Lines from 62 to 80 from Introduction seem part of the results, and from 81-84 part of conclusions

• We recognize that the Introduction section of a research article does not always preview the study’s results and contributions; however, this is a standard practice in the research communities to which we belong (and in which we hope our work will be read). We are, however, open to omitting these sentences if this is strongly preferred.

I have missed some tables to tabulate the qualitative analysis, it would easier to follow the results by tabulating factors and comments by interview and type of participant. I recommend to do this.

• As requested, we have provided a coding table with factors by interview number as supporting information.

Data are not available, according to authors, data will be sahred only if the paper is acepted, however if the paper is accepted it has not sense to review the data. This seems cotradictory with the subject of the paper

• The repository where we are depositing our data does not have the capability to share data privately with reviewers and we prefer not to make the data publicly available until the article has been accepted, an approach that is consistent with PLOS policy.

• If the reviewer does wish to review the data, then we can work with PLOS ONE to provide the data in a way that maintains the integrity of the blinded review process (i.e., we cannot send it directly) and does not result in duplicate publication of the data as supporting information.

---

## [Decision Letter · Decision Letter 1]

29 Apr 2022

PONE-D-21-21899R1Seeing oneself as a data reuser: How subjectification activates the drivers of data reuse in sciencePLOS ONE

Dear Dr. LaFlamme,

Thank you for submitting your manuscript to PLOS ONE. After careful consideration, we feel that it has merit but does not fully meet PLOS ONE’s publication criteria as it currently stands. Therefore, we invite you to submit a revised version of the manuscript that addresses the minor issues raised by reviewer 2.

We look forward to receiving your revised manuscript.

Kind regards,

Sergi Fàbregues

Academic Editor

PLOS ONE

Journal Requirements:

Reviewers' comments:

Reviewer's Responses to Questions

**Comments to the Author**

1. If the authors have adequately addressed your comments raised in a previous round of review and you feel that this manuscript is now acceptable for publication, you may indicate that here to bypass the “Comments to the Author” section, enter your conflict of interest statement in the “Confidential to Editor” section, and submit your "Accept" recommendation.

Reviewer #2: All comments have been addressed

Reviewer #3: All comments have been addressed

2. Is the manuscript technically sound, and do the data support the conclusions?

Reviewer #2: Yes

Reviewer #3: Yes

3. Has the statistical analysis been performed appropriately and rigorously? 

Reviewer #2: Yes

Reviewer #3: N/A

4. Have the authors made all data underlying the findings in their manuscript fully available?

Reviewer #2: Yes

Reviewer #3: (No Response)

5. Is the manuscript presented in an intelligible fashion and written in standard English?

Reviewer #2: Yes

Reviewer #3: Yes

6. Review Comments to the Author

Reviewer #2: It's wonderful to see that authors have made some effective and clear changes. But I still have some questions and expectations that could be better.

The article takes the cross-level relationship between influencing factors as an important innovation of this paper, rather than the discovery of factors (in section “discussion”). Based on the review and participant descriptions, I assume that this cross-level relationship is one in which a factor exerts its influence across both institutional and individual levels, correct? If so, please describe it clearly. If not, please also add what levels it is across. The current description makes it easy to assume that a factor is acting across both project-independent and project-independent dimensions (although the authors clearly state this in their reply comments).

Furthermore, do all factors have cross-level relationships? Or only some of them? What is the role played by the same factor across levels? The current results section mentions this situation and expresses this relationship in a rather fragmented and implicit formulation, which needs to be explored in depth in the discussion.

Reviewer #3: (No Response)

7. PLOS authors have the option to publish the peer review history of their article (what does this mean?). If published, this will include your full peer review and any attached files.

Reviewer #2: No

Reviewer #3: No

---

## [Author Response · Author response to Decision Letter 1]

9 Jun 2022

We would like to thank the reviewer for their additional comments and suggestions. Please see our point-by-point responses below.

Reviewer #2 Comments

It's wonderful to see that authors have made some effective and clear changes.

• Thank you for these encouraging comments; we are pleased to hear that our previous round of revisions clarified the intervention that we are seeking to make.

The article takes the cross-level relationship between influencing factors as an important innovation of this paper, rather than the discovery of factors. Based on the review and participant descriptions, I assume that this cross-level relationship is one in which a factor exerts its influence across both institutional and individual levels, correct? If so, please describe it clearly. If not, please also add what levels it is across. The current description makes it easy to assume that a factor is acting across both project-independent and project-independent dimensions (although the authors clearly state this in their reply comments).

• Thank you for pointing to the ambiguity in our use of the term “cross-level relationship.” To resolve this, we have introduced a distinction between factor type and level of analysis. Each of the seven influencing factors is associated with one of two factor types: project-independent or project-dependent. Each of the seven factors is also associated with one of three levels of analysis: individual, project, or institutional level. We have updated our model (Fig 1) to reflect this distinction, and have specified when we are referring to factor type vs. level of analysis throughout the manuscript.

Furthermore, do all factors have cross-level relationships? Or only some of them? What is the role played by the same factor across levels? The current results section mentions this situation and expresses this relationship in a rather fragmented and implicit formulation, which needs to be explored in depth in the discussion.

• Thank you for these insightful questions, which helped us to recognize not only some linguistic ambiguity but also conceptual imprecision in our use of the term “cross-level relationship.” For the sake of clarity, this term no longer appears in our manuscript. Our position is not that individual factors have cross-level relationships (although our data show that some project-level factors, including trust in the data and capabilities to reuse, are shaped by both individual and institutional conditions). Rather, our position is that all seven factors, taken together, exert an influence on recurring reuse behavior across levels of analysis, which we are able to explain by reference to the activating mechanism of subjectification as an inherently multilevel process. We have spelled out this argument explicitly in the revised second paragraph of the Discussion section (lines 764-783).

---

## [Decision Letter · Decision Letter 2]

14 Jul 2022

Seeing oneself as a data reuser: How subjectification activates the drivers of data reuse in science

PONE-D-21-21899R2

Dear Dr. LaFlamme,

We are pleased to inform you that your manuscript has been judged scientifically suitable for publication and will be formally accepted for publication once it meets all outstanding technical requirements.

Kind regards,

Sergi Fàbregues

Academic Editor

PLOS ONE

Reviewers' comments:

Reviewer's Responses to Questions

**Comments to the Author**

1. If the authors have adequately addressed your comments raised in a previous round of review and you feel that this manuscript is now acceptable for publication, you may indicate that here to bypass the “Comments to the Author” section, enter your conflict of interest statement in the “Confidential to Editor” section, and submit your "Accept" recommendation.

Reviewer #2: All comments have been addressed

2. Is the manuscript technically sound, and do the data support the conclusions?

Reviewer #2: Yes

3. Has the statistical analysis been performed appropriately and rigorously? 

Reviewer #2: Yes

4. Have the authors made all data underlying the findings in their manuscript fully available?

Reviewer #2: Yes

5. Is the manuscript presented in an intelligible fashion and written in standard English?

Reviewer #2: Yes

6. Review Comments to the Author

Reviewer #2: The authors have done a very clear job of explaining the subject and have addressed my previous doubts very well.

7. PLOS authors have the option to publish the peer review history of their article (what does this mean?). If published, this will include your full peer review and any attached files.

Reviewer #2: No

---

## [Editor Report · Acceptance letter]

10 Aug 2022

PONE-D-21-21899R2 

Seeing oneself as a data reuser:How subjectification activates the drivers of data reuse in science 

Dear Dr. LaFlamme:

I'm pleased to inform you that your manuscript has been deemed suitable for publication in PLOS ONE. Congratulations! Your manuscript is now with our production department. 

Kind regards, 

on behalf of

Dr. Sergi Fàbregues 

Academic Editor

PLOS ONE